# Intelligent Localization Sampling System Based on Deep Learning and Image Processing Technology

**DOI:** 10.3390/s22052021

**Published:** 2022-03-04

**Authors:** Shengxian Yi, Zhongjiong Yang, Liqiang Zhou, Shaoxin Zou, Huangxin Xie

**Affiliations:** State Key Laboratory of High-Performance Complex Manufacturing, School of Mechanical and Electrical Engineering, Central South University, Changsha 410083, China; shengxian21@126.com (S.Y.); csurobert@csu.edu.cn (L.Z.); shaoxin0108@126.com (S.Z.); xiehuangxincqu@163.com (H.X.)

**Keywords:** visual localization, deep learning, image processing, SSD, intelligent sampling robot, area sensing, automatic sampling of mineral powder

## Abstract

In this paper, deep learning and image processing technologies are combined, and an automatic sampling robot is proposed that can completely replace the manual method in the three-dimensional space when used for the autonomous location of sampling points. It can also achieve good localization accuracy, which solves the problems of the high labor intensity, low efficiency, and poor scientific accuracy of the manual sampling of mineral powder. To improve localization accuracy and eliminate non-linear image distortion due to wide-angle lenses, distortion correction was applied to the captured images. We solved the problem of low detection accuracy in some scenes of Single Shot MultiBox Detector (SSD) through data augmentation. A visual localization model has been established, and the image coordinates of the sampling point have been determined through color screening, image segmentation, and connected body feature screening, while coordinate conversion has been performed to complete the spatial localization of the sampling point, guiding the robot in performing accurate sampling. Field experiments were conducted to validate the intelligent sampling robot, which showed that the maximum visual positioning error of the robot is 36 mm in the x-direction and 24 mm in the y-direction, both of which meet the error range of less than or equal to 50 mm, and could meet the technical standards and requirements of industrial sampling localization accuracy.

## 1. Introduction

In the fiercely competitive environment of the metallurgical industry, the efficiency of ore powder quality inspection is directly related to the costs and benefits of the enterprise. As shown in Figure 1, enterprises must randomly sample several areas of the transport carriage, and manual inspections, with high labor intensity and low operating efficiency, will directly expose the inspector to the risk of heavy metal pollution. If automated robots are used to sample, it can not only reduce labor costs and improve efficiency, but also avoid the risk of corruption; therefore, automation and intelligence are obviously important directions for the future development of the mineral and industrial fields [1,2,3]. However, many of the currently used mine powder sampling machines have shortcomings, such as their small sampling range, insufficient intelligent working ability, long sampling and localization time, and insufficient sampling localization accuracy. Traditional conventional sampling techniques can no longer meet the needs of modernization [4,5,6,7,8].

In recent years, target detection algorithms based on deep learning have been widely used in various detection scenarios [9], mainly because of their strong comprehensiveness, activity, and ability to detect and recognize multiple types of objects at the same time [10,11,12,13]. They can be roughly divided into two categories—two-stage algorithms based on the candidate area, such as the R-CNN series, fast R-CNN, faster R-CNN, etc. [14,15,16], and one-stage algorithms based on the regression method, such as YOLO (You Only Look Once) series and SSD [17,18,19,20,21]. Two-stage algorithms have high accuracy but slow speed, and are not suitable for the real-time requirements of ore powder target detection. One-stage algorithms have higher speed, better real-time performance in target detection, and their accuracy can reach the level of FASTER R-CNN [22]. Therefore, the one-stage algorithm is more suitable for the target detection of ore powder than the two-stage algorithm.

The SSD algorithm and YOLO algorithm, as typical representatives of one-stage algorithms, are widely used in the engineering field [23]. However, the YOLO algorithm also has certain limitations. The YOLO algorithm has a weak generalization ability for object aspect ratio, and its detection accuracy decreases when the aspect ratio of a new class of objects appears [24,25,26]. SSD has two advantages, namely, real-time processing and high accuracy, especially when targeting objects in different size scales, meaning a certain level of accuracy can be guaranteed [27,28,29,30,31]. Given that the size of the ore hauling vehicle to be detected is generally variable and uncertain, the SSD algorithm is comparatively more suitable for the multi-scale detection of ore hauling vehicles. Thus, given the detection task of this study, the SSD algorithm was selected as the detection model for the sampling area.

However, SSD has the limitation that small objects are not detected well. This is not only a problem for SSD, but also for most object detection algorithms. To solve this problem, various attempts have been made at replacing VGGNet with ResNet [32,33] or feature fusion [34]. Fu et al. proposed another version of SSD called Deconvolutional SSD (DSSD), which used ResNet instead of VGGNet and achieved better performance compared to the conventional SSD; however, DSSD improved accuracy at the cost of speed [35]. Jeong et al. improved the method of feature fusion, so as to make full use of the features of each output layer [36]. Li et al. proposed the feature fusion Single-Shot MultiBox Detector (FSSD) model, which obtained more details of the output feature layers through feature fusion and down-sampling, so as to improve the detection accuracy of the model [37]. Although the improved SSD algorithm can increase the detection accuracy, the automatic detection of mineral powder involves not only detecting the area of the carriage, but also detecting and locating the 3 × 3-pixel locating light spot in the image. The current SSD algorithm cannot achieve such accuracy, even though it is improved temporarily, and it cannot achieve direct detection, so it needs to be integrated with other image technologies to enable automated localization and sampling detection.

Currently, image processing techniques, such as color screening and image segmentation, are widely used and studied. In terms of HSV, Qussay Al-Jubouri et al. proposed a method for classifying individual zebrafish based on statistical texture and hue/saturation/value (HSV) color features. Experimental comparisons show that the effectiveness of this method is significantly improved compared with previous individual classification methods [38]. Tomáš Harasthy et al. proposed a traffic sign detector based on the HSV color model, and the success rate of the detector exceeded 95% when using the HSV color model [39]. Dede Wand et al. applied a fusion of HSV and HSI methods to detect the wilting condition of roses, and concluded that this method was the fastest [40]. K. Cantrell et al. used the hue parameters of hue, saturation, and value color space as quantitative analytical parameters for bitonal optical sensors. Studies have shown that the H value can maintain excellent accuracy even under changes of indicator concentration, film thickness, and detector spectrum, showing its great prospects for application in various sensing applications [41]. In addition, in terms of image segmentation, Wang et al. introduced a new automatic Region-based Image Segmentation Algorithm based on k-means clustering (RISA), specifically designed for remote sensing applications. This method was shown to have better flexibility and accuracy through case studies [42]. Silvia Ojeda et al. introduced a new algorithm to perform image segmentation; the foundations of this algorithm are random field theory and its robustness when used for spatial autoregressive processes. Experimental results with real images have verified how the algorithm works in practice [43]. Siddharth Singh Chouhan et al. investigated the existing application of Soft Computing techniques in image segmentation, and their research showed that the use of Soft Computing methods can deal with uncertainty and inaccuracy, thus enabling intelligent systems to gain flexibility and adaptability [44]. According to the current research, image processing technology is widely used in various fields related to images owing to its superiority in real time, its adaptability, and its high accuracy. In this study, the SSD algorithm based on deep learning is combined with image processing technologies such as color screening and image segmentation to achieve accurate localization and coordinate transformation of sampling points, so as to guide the robot to conduct automatic detection and meet the requirements of the error range of localization. Compared with the traditional conventional sampling technology, it has the advantages of shorter time-consuming sampling and positioning, stronger intelligent working ability, and larger sampling range.

The main contributions of this study are as follows:(1)Using the checkerboard calibration board and the MATLAB toolbox, the camera’s internal parameters and the distortion parameters have been determined. Then, the acquired images were distortion-corrected by OpenCV (Open Source Computer Vision Library) to improve the accuracy of sampling area detection and visual localization. The use of data augmentation has solved the problem of the lower availability of data sets of the same vehicle in different environments and weather conditions, as well as that of the low accuracy of the SSD algorithm when used in detecting compartments in some scenes;(2)The visual localization model was established, and the coordinates of the sampling point were located by color screening, image segmentation, and connecting body feature screening. The coordinates of the sampling point were converted from the image coordinate system (ICS) to the sampling plane coordinate system (SPCS), and then the robot was guided to the corresponding random point for accurate localization and sampling;(3)A sampling verification experiment was carried out on the delivery truck on the spot. The experimental results showed that the visual localization errors of the robot in the x-direction and y-direction were less than the error margins, which means it could meet the accuracy requirements of on-site localization and sampling, and facilitate automatic sampling by a robot.

The flow chart of the study system is as follows:



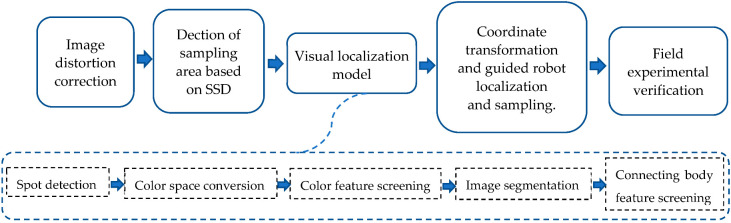



## 2. The Proposed System

In this section, the proposed system is explained in detail. The proposed system is based on four sections: image distortion correction; SSD detection of the area of the carriage; the visual localization model; coordinate transformation, and guided robot localization and sampling. The details are shown below.

### 2.1. Distortion Correction

#### 2.1.1. Lens Distortion

The ore powder transport vehicles kept at the detection site are of different sizes, and the carriages of the largest car are six to eight meters long. As shown in the Figure 2 below, the height of the pound room containing the quality inspection area is eight meters. Because the coverage of the scene to be shot is relatively large, the whole area cannot be shot with an ordinary lens, so a wide-angle lens must be adopted.

However, due to the optical system design, processing and manufacturing processes, external forces and other factors, a wide-angle lens will cause nonlinear distortion in the image captured; hence, the accurate transformation of the coordinate system cannot be achieved by using the ideal pinhole imaging model. Therefore, before the sampling area is detected in the image, the distortion parameters need to be calculated via camera calibration so as to correct the distortion of the image of the site.

Thin lens distortion refers to tilt distortion in the tangential and radial directions caused by the design of the lens itself, deviations in lens manufacturing, and errors in the processing and assembling. In general, thin lens distortion has little influence on the system and can be ignored; only radial distortion and centrifugal distortion are considered. Image distortion can be expressed as follows [45,46]:(1)$$\left[\begin{array}{ccccc} {cR}^{2} & {cR}^{4} & {cR}^{6} & 2cr & R^{2}{+2c}^{2}\\ {rR}^{2} & {rR}^{4} & {rR}^{6} & R^{2}{+2r}^{2} & 2cr \end{array}\right]\left[\begin{array}{c} k_{1}\\ k_{2}\\ k_{3}\\ p_{1}\\ p_{2} \end{array}\right]=\left[\begin{array}{c} c_{\mathrm{corrected}}-c\\ r_{\mathrm{corrected}}-r \end{array}\right]$$

Here, (c,r) is the original position of the distortion point in the ICS, and $\left(c_{corrected},r_{corrected}\right)$ is the position of the distortion point after distortion correction, considering radial distortion and centrifugal distortion.

We convert Equation (1) into DK=d, $K={\left[\begin{array}{ccccc} k_{1} & k_{2} & k_{3} & p_{1} & p_{2} \end{array}\right]}^{T}$, $D=\left[\begin{array}{ccccc} cR^{2} & cR^{4} & cR^{6} & 2cr & R^{2}+2c^{2}\\ rR^{2} & rR^{4} & rR^{6} & R^{2}+2r^{2} & 2cr\; \end{array}\right]$, and $d=\left[\begin{array}{c} c_{corrected}-c\\ r_{corrected}-r \end{array}\right]$. Then, the distortion parameters K can be obtained by the least-squares method.
(2)$$K={\left(D^{T}D\right)}^{-1}D^{T}d$$

#### 2.1.2. Camera Calibration

Because the actual internal parameters of the camera are often inconsistent with expectations, in the actual use of cameras, it is necessary to determine the accurate internal parameters [47]. As shown in Figure 3 below, a 10×7 checkerboard was selected as the calibration object for the industrial camera. The black and white grids in the checkerboard are evenly distributed, and have good stability under different light intensities, which makes the image useful for completing the operation of corner calibration.

The MATLAB camera calibration toolbox is used to obtain images with different angles and different positions by changing the position of the camera or the calibration plate. Then, the corners of the calibration plate are extracted from the image, and five internal matrix parameters and five distortion parameters are solved. The parameters are optimized through the Levenberg–Marquardt algorithm. The specific process of camera calibration is as follows [48]:

(1) Attach the printed calibration plate to a flat surface to ensure that it is flat;

(2) Images of calibration plates at different angles and positions can be obtained by moving the camera or calibration plate. The number of calibration images is 17 (generally 15–25 [49]). During the shooting process, ensure that the calibration plate is captured completely, and the position of the calibration plate covers the image as much as possible, so as to ensure enough information is captured to calculate the projection matrix of the camera, as shown in Figure 4 below.

(3) Use the MATLAB camera calibration toolbox to detect the corners in each image, and select the border of the calibration plate manually during the operation to improve the calibration accuracy;

(4) According to the actual distance between each of the two corners of the calibration plate and the corner information extracted from the image, the internal parameter matrix of the camera was calculated, and the distortion parameters of the camera were solved by the least-squares method. Then, the maximum likelihood estimation was used to optimize the parameters.

#### 2.1.3. Distortion Correction Based on MATLAB Camera Calibration Toolbox

The MATLAB camera calibration toolbox was employed to read the 17 captured images of the calibration plates, as shown in Figure 5a. The corners of the 17 images were extracted via the Extract grid corners function. In the process of corner extraction, the four corners of the calibration plate need to be manually specified. After accurate calibration of the range, the actual sizes of the black and white grids of the calibration plate need to be input manually. After the input is complete, the calibration toolbox will automatically extract the corners of the calibration plate and mark them in the image, as shown in Figure 5b.

After the corners of 17 images have been extracted, the relative position and angle of the camera and the calibration plate can be intuitively determined through the Reproject images function. As shown in Figure 6a, $O_{c}\;$on the left is the origin of the camera coordinate system, and the camera coordinate system $\left(O_{c},X_{c},Y_{c},Z_{c}\right)\;$is established. The different grids represent the positions of the calibration plate relative to the camera coordinate system. In Figure 6b, the calibration plate in the world coordinate system is used as a reference, and the different pyramids represent the positions of the camera relative to the calibration plate.

The projection error distribution of the corners extracted from images 1–17 can be obtained through the Analyze error function, as shown in Figure 7.

In the 17 calibration plate images showing different positions, the maximum value of the projection error in the x-direction is less than 1.5 pixels, and the maximum value of the projection error in the y-direction is less than 2 pixels. The error is small and meets the requirements of system accuracy.

After the camera is calibrated, the internal parameters and distortion parameters of the camera are calculated through the Calibration function. The camera internal parameters table includes scale factors $f_{u}$ and $f_{v}$ ($f_{u}=\frac{f_{x}}{du},\;f_{v}=f_{y}/dv$) in the x- and y-directions, the image center coordinates $c_{0}$ and $r_{0}$, and the distortion factor $\alpha_{c}$. The specific parameters’ values are shown in Table 1.

$k_{1},\;k_{2},\;\mathrm{and}\;k_{3}\;$are radial distortion parameters, $p_{1}\;\mathrm{and}\;p_{2}\;$are centrifugal distortion parameters, and the specific parameter values are shown in Table 2.

After the camera’s internal parameters and distortion parameters are obtained, distortion correction is carried out through the initUndistortRectifyMap function and remap function of OpenCV, and the comparison of before and after distortion correction is shown in Figure 8.

The image captured by the industrial camera shows obvious nonlinear distortion, including obvious pincushion distortion at the edge of the image, and the carriage also shows slight barrel distortion. This will not only affect the detection accuracy of the sampling area, but also the subsequent visual localization accuracy. After distortion correction, the nonlinear distortion of the image is significantly improved, which lays the foundation for the subsequent localization accuracy.

### 2.2. SSD Algorithm Is Used to Detect the Carriage Area

#### 2.2.1. SSD Algorithm

The network structure of the SSD algorithm model is shown in Figure 9. Each convolutional layer will output feature maps of the receptive fields of different sizes. On these different feature maps, the target locations and categories are trained and predicted so as to achieve multi-scale detection. This can also overcome the problems of the uncommon aspect ratio and the low recognition accuracy, and greatly improve the generalization ability [21], which is also key to the selection of SSD in this paper.

#### 2.2.2. Experimental Process and Analysis of Sampling Area Detection

##### Image Acquisition

Since the industrial camera is fixed on the top and the scene is semi-enclosed, factors such as the type of vehicle, shadow, image brightness, and others have a greater impact on the detection of the sampling area. In order to improve the adaptability of the SSD algorithm to the environment, the scene was pictured at different times. The pictures of the trucks in different weather conditions are used as the training set, and some of the pictures from the training set are shown in Figure 10 below.

##### Data Set Pre-Processing

The training of the target detection model requires a large number of data sets for feature extraction and learning. The pictures collected on site are limited. In order to obtain a data set with sufficient training samples, brightness and contrast adjustment, noise addition, image flipping, image blurring, and other operations are performed on each picture via data augmentation [50], as shown in Figure 11 below.

##### Data Set Annotation

Before model training, the sampling range in each training set image needed to be labeled with a real frame. LabelImg was used to label each image in each training set with a rectangular frame, as shown in Figure 12 below. After labeling, an XML file was generated to store the target category of the image and the position information of the rectangular frame, etc. During the training, the vector parameters in the XML file were used as the default frame in the regression process.

##### Data Conversion and Division

The deep learning framework used in the experiment is Tensorflow-GPU-1.12, which performs the conversion from XML format to tfrecord format through script files. In the process of data conversion and storage, samples in the data set are divided into a training set and a verification set (3:1), as shown in Figure 13 below.

#### 2.2.3. Training Process

Before data training, the parameters related to the model training process should be set, as shown in Table 3.

Here, weight_decay is the weight decay coefficient; optimizer is the optimizer of the model’s gradient descent; initial_learning_rate is the initial learning rate, and decay_factor is the decay factor. When approaching the optimal solution, the convergence speed of the model can be accelerated by reducing the learning rate. Batch_size is the size of each batch of data, and num_steps is the number of training steps.

The image size of the training set is 819 × 614 pixels, and the number of training cycles is 20,000, which takes 5 h and 36 min. During the training process, the change of classification loss is as shown in Figure 14a, the change of localization loss is as shown in Figure 14b, and the change of total loss is as shown in Figure 14c. With the increase in training cycles, the loss gradually converged.

After training, a model is generated that can be called. Through the generated model, the location of the sampling area of the field image can be detected, and a prediction frame can be generated.

#### 2.2.4. Regional Detection Results and Analysis

After training, the model was used in the test set for detection, and the detection effects before and after data augmentation were compared. If the prediction frame generated by the SSD algorithm exceeds the actual sampling range, this will cause security problems. If the generated prediction frame is too small, it will not completely cover all sampling areas. Therefore, $IoU_{G}$ is defined as the ratio of the area of the intersection between the prediction frame and the real frame, divided by the area of the real frame. $IoU_{G}$ is used to measure the detection effect of the SSD algorithm in the sampling area. If the prediction frame exceeds the sampling range, the sampling range is not detected or $IoU_{G}\;$< 0.8, at which point the detection is considered a failure; otherwise, the detection is considered to be successful. The trained model performs sampling area detection on 200 test sets, and the detection results are shown in Figure 15 below.

As can be seen from Table 4, before data augmentation, due to the time limit of transport vehicles, there are fewer data sets available for the same vehicle in different light intensities and contrasts, which results in two test images not within the sampling range of image detection. The prediction frame of 25 test set images exceeds the sampling range, and the prediction frame of 23 test set images $IoU_{G}\;$< 0.8. After data augmentation, the detection success rate increases from 74% to 92%, indicating that using data augmentation can effectively improve the detection success rate, as well as solving the problem of the low detection accuracy of some scenes using SSD (Single-Shot MultiBox Detector).

### 2.3. Visual Localization Model

Due to the employed plane of sampling and the plane parallel to the camera coordinate system, conversion between the image plane coordinate system and the world coordinate system can be simplified. A visual localization model based on the bridge sampling robot, which is easy to deploy and meets the requirements of sampling accuracy, is proposed. Taking the centroid of the laser spot in the image as the reference, and combining this with the distance of the camera and the laser localization lamp, the sampling points generated in the image are translated into the SPCS, which is used to guide the sampling robot towards the randomly generated sampling points. The schematic diagram of the visual localization model is shown in Figure 16.

In the visual localization model, the conversion from the ICS to the SPCS can be achieved through the following steps.

(1)After the laser localization lamp on the top projects the green spot onto the sampling plane, through the imaging characteristics of the laser spot, it can determine the position of the green laser spot in the image, and then calculate the distance Δc between the center point of the image and the centroid of the spot.(2)Based on the installation distance between the industrial camera and the laser localization lamp (∆x and ∆c), the factor of conversion K_t_ from the ICS to the SPCS is calculated, and then the conversion from the sampling point coordinates (c_i_, r_i_) of the ICS to the actual sampling point coordinates (x_i_, y_i_) of the SPCS is performed.

#### 2.3.1. Spot Detection

The laser spot has good directivity and small divergence, and is commonly used in optical localization and optical measurement [51]. As shown in Figure 17 below, whether it is cloudy or sunny, there is a large difference between the laser spot and the background. Therefore, the difference between the laser spot and the background can be used in combination with the characteristics of the laser spot to effectively detect the laser spot.

#### 2.3.2. Color Space Conversion

Although the color pictures stored and displayed on a computer are usually in RGB color space, RGB color space is more sensitive to light, and the color is greatly affected by brightness. Since HSV color space is more in line with human visual characteristics, the changing of various color components can be more intuitively felt in HSV color space. Moreover, in the H channel (hue channel), the color distribution is a continuous interval. Therefore, depending on the laser spot’s color, a range of hue intervals can be set to convert the image from the RGB color space to the HSV color space so as to reduce the influence of light on the detection of green laser spots. The color space conversion process is shown in Figure 18.

Both HSV and RGB use three channels to describe color information, so there is a linear transformation between the three channels of RGB and HSV. For conversion from RGB color space to HSV color space, the values of the R, G, and B channels are normalized first, the value range of each is set to [0, 1], the value range of channel H is set to [0°, 360°], and the value range of channel S and Channel V is set to [0, 1]. Color space conversion is carried out through the following equation [52]:(3)$$H=\{\begin{array}{c} \;0{}^{\circ}\;,if\;max=min\\ \;60{}^{\circ}\times \frac{G-B}{max-min}+0{}^{\circ}\;,if\;max=R\;and\;G\geq B\\ \;60{}^{\circ}\times \frac{G-B}{max-min}+360{}^{\circ},if\;max=R\;and\;G<B\\ 60{}^{\circ}\times \frac{B-R}{max-min}+120{}^{\circ},if\;max=G\\ 60{}^{\circ}\times \frac{R-G}{max-min}+240{}^{\circ},if\;max=B \end{array}$$
(4)$$S=\{\begin{array}{c} \;0,\;if\;max=0\;\\ \frac{max-min}{max}=1-\frac{min}{max},otherwise \end{array}$$
(5)V=max 
where *H* is the hue value of the pixel in the HSV color space, *S* is the saturation value of the pixel in the HSV color space, *V* is the brightness value of the pixel in the HSV color space, *R* is the red component of pixels in the RGB color space, *G* is the green component of pixels in the RGB color space, and *B* is the blue component of the pixel in the RGB color space. Max represents the maximum value of *R*, *G*, and *B*, and min represents the minimum value of *R*, *G*, and *B*.

Through the above equations, the color conversion from the RGB color space to the HSV space can be completed, and then, pixels of corresponding colors can be screened from the original image according to the actual imaging color of the laser spot.

#### 2.3.3. Color Feature Screening

Since the image is stored in the computer, the computer compresses the data into one byte, in order to facilitate processing. Therefore, in OpenCV, the range of values for the three channels in the RGB and HSV images is 0~255.

Based on the color of the green laser projected onto the mineral powder plane, we set the ranges of the H channel, S channel, and V channel to ensure that the green pixels in the image can be accurately detected in the image under different lighting conditions. Based on the HSV color space model, the specific settings are shown in Table 5.

Based on the setting range of the three HSV channels, a mask (object template) is generated, and a logical AND operation is performed with the original image to obtain all pixels that meet the conditions in the HSV image, as shown in Figure 19.

From the partial enlargement shown on the right, the green laser spot can be screened from the HSV image based on the color characteristics, but the other green pixels in the image may be screened as well. The positions of these interfering pixels are random, so the position of the green laser spot cannot be obtained directly from the image using color features alone. It is also necessary to screen out the position of the green laser spot based on other features of the green laser spot after color screening.

#### 2.3.4. Image Segmentation

Although green pixels can be screened out from the image based on the HSV color space, there are also green interference pixels that affect the detection of green laser spots, so the image segmentation method is adopted to separate the green laser spots from the background.

The most common methods of image segmentation include the region growing method and threshold segmentation method. In the research, the area of the green laser spot in the image is small, and the position in the image also changes with the height of the sampling plane, which is not conducive to the setting of seed pixels. In addition, the noise or uneven grayscale distribution of the image may cause voids or over-segmentation, so it is not suitable for image segmentation using the region growing method; as such, the relatively stable threshold segmentation method is used to extract the green laser spot.

Threshold segmentation methods mainly include the artificial threshold method and the maximum variance method. Whether the artificial threshold method can effectively segment the image mainly depends on whether there is enough contrast between the object and the background. However, under different lighting conditions, there are differences in the images captured by the industrial camera. The artificial threshold method cannot effectively separate the green laser spot from the background, so the maximum interclass variance method is used instead.

Assuming that the image consists of only the target object and the background, the grayscale distribution probability density of the background is $F_{1}\left(T\right)$, the grayscale distribution probability density of the target object is $F_{2}\left(T\right)$, and the ratio of the pixel of the target object to the pixel of the whole image is φ, so the grayscale probability density distribution F(T) of the whole image can be expressed through the following equation [53]:(6)$$F\left(T\right)=\varphi F_{2}\left(T\right)+\left(1-\varphi \right)F_{1}\left(T\right)$$

Assuming that the image consists of a dark background and a bright target object, $F_{1}\left(T\right)\;$and $F_{2}\left(T\right)$ can be represented in Figure 20. Pixels with grayscale values less than $T_{z}$ are considered as the background, and pixels with grayscale values greater than $T_{z}$ are considered as the target object [54].

After segmentation of the background and the target object via gray threshold $T_{z}$, the probability of the background being mistakenly used as the target object can be calculated as [55]:(7)$$E_{1}\left(T_{z}\right)=\int_{-\infty }^{T_{z}}F_{1}\left(T\right)dT$$

Similarly, the probability of the target object being mistakenly used as the background is:(8)$$E_{2}\left(T_{z}\right)=\int_{-\infty }^{T_{z}}F_{2}\left(T\right)dT$$

The total probability of mis-segmentation is the sum of $E_{1}\left(T_{z}\right)$ and $E_{2}\left(T_{z}\right):$(9)$$E\left(T_{z}\right)=\left(1-\varphi \right)E_{1}\left(T_{z}\right)+\varphi E_{2}\left(T_{z}\right)$$

If there is a $T_{z}$ that can make $E\left(T_{z}\right)\;$obtain the minimum value, then $T_{z}$ is the optimal threshold. The optimal threshold $T_{z}$ can be calculated by deriving $E\left(T_{z}\right)$ and making $\frac{dE\left(T_{z}\right)}{dT_{z}}=0$; the result is as follows:(10)$$\varphi F_{2}\left(T_{z}\right)=\left(1-\varphi \right)F_{1}\left(T_{z}\right)$$

Assuming that $F_{1}\left(T\right)$ and$\;F_{2}\left(T\right)$ satisfy a normal distribution [56], the mean grayscale values are $\mu_{1}$ and $\mu_{2}$, and the standard deviations are $\sigma_{1}$ and $\sigma_{2}$, $F_{1}\left(T\right)$ and$\;F_{2}\left(T\right)\;$can be expressed through the following equation:(11)$$F_{1}\left(T\right)=\frac{1}{\sqrt{2\pi }\sigma_{1}}exp\left[\frac{-{\left(T-\mu_{1}\right)}^{2}}{2\sigma_{1}{}^{2}}\right]$$
(12)$$F_{2}\left(T\right)=\frac{1}{\sqrt{2\pi }\sigma_{2}}exp\left[\frac{-{\left(T-\mu_{2}\right)}^{2}}{2\sigma_{2}{}^{2}}\right]$$

Substituting (11) and (12) into (10), and taking the logarithm of both sides of the equation and simplifying them, the following equation can be obtained:(13)$$AT_{z}{}^{2}+BT_{z}+C=0$$

Here, $A=\sigma_{1}{}^{2}-\sigma_{2}{}^{2}$, $B=2\left(\mu_{1}\sigma_{1}{}^{2}-\mu_{2}\sigma_{2}{}^{2}\right)$, and $C=2\sigma_{1}{}^{2}\sigma_{2}{}^{2}ln\left(\sigma_{2}\varphi /\sigma_{2}\left(1-\varphi \right)\right)$$+\sigma_{1}{}^{2}\mu_{2}{}^{2}-\sigma_{2}{}^{2}\mu_{1}{}^{2}$.

Based on the information in the image, the parameters $\varphi ,\;\mu_{1},\;\sigma_{1},\;\mu_{2},\;\mathrm{and}\;\sigma_{2}\;$are solved. Then Equation (13) is solved, and the result of the solution is the optimal threshold value.

After segmentation of the grayscale image through the maximum interclass variance method, a binary image is obtained. The foreground of the binary image contains not only the connecting body of the green laser spot, but also the connecting body of the interfering pixel, as shown in Figure 21b:

The binary image contains the connecting body of the interfering pixels, so it is necessary to screen using the connecting body characteristics of the green laser spot to obtain its centroid position.

#### 2.3.5. Connecting Body Feature Screening

Based on the features of laser spot imaging, a total of three contiguous features, namely, the center position (x_M_, y_M_), the contiguous area feature S, and the contiguous circularity feature θ, were selected for screening [57].
(14)$$\{\begin{array}{c} c_{min}\leq c_{Mi}\leq c_{max}\;\\ r_{min}\leq r_{Mi}\leq r_{max}\\ \theta_{min}\leq \theta_{i}\leq 1\\ S_{min}\leq S_{i}\leq S_{max} \end{array}$$
where *L* is the long axis of the continuum; *i* is the number of the continuum in the binarized image; *n* is the total number of the continuum in the binarized image; $c_{min}\;\mathrm{and}\;c_{max}$ are the limit positions of the green laser spot’s center coordinates in the c-direction; $r_{min}\;\mathrm{and}\;r_{max}$ are the limit positions of the green laser spot’s center coordinates in the direction of r; $c_{min},\;c_{max},\;r_{min},\;\mathrm{and}\;r_{max}$ can be measured in the lowest sampling plane and the highest sampling plane; $S_{min}\;\mathrm{and}\;S_{max}$ are the limit area of the green laser spot’s continuum; $\theta_{min}$ represents the minimum value of roundness of the green laser spot’s continuum, and $S_{min},\;S_{max},\;\mathrm{and}\;\theta_{min}$ can be set according to the field environment.

Through the above-described feature screening, the green laser spot can be detected in the image, the center coordinates of the green spot are output, and the detection results are marked out (with the center coordinates shown as the circle). The detection results of the green laser spot in different scenes corresponding to the image are shown in Figure 22.

All images taken in the field have been detected, and the center position of the green laser spot irradiated on the sampling plane has been accurately located in the image, providing accurate information for the subsequent conversion of the ICS to the sampling plane.

### 2.4. Coordinate Transform

Since the sampling depth in the z-direction of the sampling robot is controlled by the motor, and the green laser spot is projected vertically on the sampling plane, the basic coordinate system of the three-dimensional sampling robot can be simplified to a two-dimensional SPCS. The origin of the SPCS coincides with the projection point of the laser localization lamp in the sampling plane. Therefore, the coordinates of the sampling point in the SPCS can be sent to the sampling robot to guide it to the corresponding location for sampling.

The lateral distance between the center of the camera and the center of the laser localization lamp can be obtained by measurement, and in the ICS, the lateral distance between the center point of the picture and the green laser spot can also be calculated. As such, the conversion from the picture coordinate system to the SPCS can be simplified to a side conversion of two similar triangles, as shown in Figure 23.

The coordinate transformation of the sampling points is calculated as follows:(15)$$K_{t}=\Delta c/\Delta x\;$$
(16)$$\left(\begin{array}{c} x_{i}\\ y_{i} \end{array}\right)=K_{t}\left(\begin{array}{c} c_{i}\\ r_{i} \end{array}\right)\;i=1,2,\ldots ,16\;$$
where ∆*c* is a constant in the SPCS, in pixels; ∆*x* varies with the height of the sampling plane, and the higher the sampling plane, the larger ∆*x*, measured in mm; $K_{t}$ is the conversion coefficient from the ICS to the SPCS, in pixel/mm, which is related to the height of the sampling plane; *i* is the serial number of the sampling point; ${\left(x_{i},\;y_{i}\right)}^{T}$ shows the sampling points in the SPCS, and ${\left(c_{i},\;r_{i}\right)}^{T}$ shows the randomly generated points in the ICS, according to the rules.

When sampling planes of different heights, Equations (15) and (16) can realize the conversion from the ICS to SPCS. The coordinates of the random sampling points can be converted and transmitted to the robot through the system, and then, the sampling robot is guided to the corresponding coordinate position for sampling.

## 3. Field Experiment Verification

The effectivity of the image acquisition and localization system, and the accuracy of visual localization, are verified through field experiments. The specific flow of the field experiment is shown in Figure 24.

During the experiment, the coordinates of the center of mass of the green laser spot in Figure 24d are (164,272), and the conversion coefficient $K_{t}=11.9282$. According to the visual localization model, the origin of the ICS needs to be changed to the center of mass coordinates of the green laser spot; then, the coordinates are converted by $K_{t}$, and finally, the obtained coordinates of the sampling point are sent to the system. In order to avoid accidental errors in the data, several sampling point localization experiments were conducted on the site using different trucks, and a total of 32 sampling points were selected for comparative analysis. Some of the generated sampling points in the ICS (the coordinate origin is the center of mass of the green laser spot) and the results after coordinate transformation (the coordinate origin is the projection point of the green laser light in the sampling plane) are shown in Table 6 below.

The difference between the actual sampling point $\left(x_{i}^{t},\;y_{i}^{t}\right)$ and the localized sampling point $\left(x_{i},\;y_{i}\right)$ is due to the image distortion caused by lens distortion, the accumulated error during the operation of the sampling robot, and the measurement error due to the flatness of the sampling plane. If the error is too large, the sampling robot may go beyond the sampling area, resulting in the failure of localization sampling, so it is necessary to verify the accuracy of the image acquisition and localization system through field experiments. After completing the sampling, the brazing rod will leave a concave column in the plane of mineral powder, as shown in Figure 25. By measuring the distance (in pixels) in the image between the center point of the concave column and the corresponding sampling point, combined with the conversion coefficient, the absolute error of each sampling point can be calculated, so the absolute error of the 16 points in Table 6 is shown in Table 7.

Since image distortion correction is the means to ensure the accuracy of visual localization and has a great impact on the accuracy of visual localization, the error before and after distortion correction is compared, and the analysis based on the sample point data from the experiment is shown in Figure 26 below.

As can be seen from the above figure, the maximum error and the average error of coordinate transformation in the x-direction and the y-direction are significantly reduced after distortion correction. The maximum error is reduced the most, with the maximum error in the x-direction reduced by 36 mm and the maximum error in the y-direction reduced by 21 mm. The experimental results verify the necessity and correctness of distortion correction in the image acquisition stage, and also contribute to the improvement of detection accuracy.

In order to facilitate the comparison of the errors between the points in the SPCS and the coordinates of the actual sampling points, the absolute error diagrams of 32 sampling points were drawn, according to Table 7 and as shown in Figure 27 below.

Due to the inaccurate localization of the sampling robot in the sampling process, resulting in sampling points being identified beyond the sampling area, according to the sampling point error range requirements, the randomly generated sampling points have a 152.78 mm error margin in the x-direction and an 83.33 mm error margin in the y-direction from the boundary of the sampling area, as shown in Table 8.

Here, the margin of error represents the maximum error of all randomly sampled points in the x- and y-directions that do not exceed the sampling area after the coordinate transformation. As can be seen from Figure 27, the maximum error in the x-direction is 36 mm, and the maximum error in the y-direction is 24 mm. The maximum errors in the x-direction and y-direction are much smaller than the error margin, so there will be no situation where the sampling location point is out of the sampling range. This also ensures that the error between the actual coordinates of the sampling point and the localization coordinates is less than or equal to 50 mm, which proves that the studied visual localization system can meet the requirements of industrialized practical use, and is feasible.

## 4. Conclusions and Further Work

### 4.1. Conclusions

The main focus of this paper is the perceptual localization of a sampling area. The spatial location information of the sampling point is obtained from the collected images by fusing the SSD algorithm with image processing techniques, and the sampling robot is provided with accurate localization coordinates of the sampling point. The main research work and results in this paper are summarized as follows:(1)The lens’ internal parameters and distortion parameters were obtained using the MATLAB toolbox and a calibration plate, and OpenCV is used to correct the distortion of the picture, eliminating the influence of image distortion caused by the industrial wide-angle lens. This ensures the accuracy of localization and detection;(2)By means of data augmentation, a large number of training sets within different scenes were obtained, and experiments were conducted on the test set. The accuracy of the experimental results reached over 92%, which solves the problem whereby some scenes of the SSD algorithm were not detected accurately, and ensures the accuracy of the SSD detection area;(3)A vision localization model for bridge sampling robots has been constructed, which can precisely locate the sampling point, and convert the coordinates of the localization point from the ICS to the base coordinate system of the sampling robot, before guiding the sampling robot to the corresponding sampling point for accurate localization and sampling;(4)The maximum error of the localization sampling system in the x-direction is 36 mm and the maximum error in the y-direction is 24 mm. This visual localization error is much smaller than the error margin in the corresponding direction, and the localization accuracy meets the error range requirement of industrialized mineral powder sampling. Compared with the traditional conventional sampling technology, it has the advantages of shorter sampling and positioning time (originally 3–5 min for a single point, but now 8 min for a total of 16 points), stronger intelligent working ability and larger sampling range (originally the sampler can only detect half of the carriage range, but now the full range can be detected). As such, the automation of robot mineral powder localization and sampling is successfully realized.

### 4.2. Further Work

The data set used in the SSD model was collected on-site within a period of three months, and lacks data from other time periods. Although data augmentation has been performed to improve accuracy, the model may still fail to locate the sampling area effectively in some future weather scenarios. Therefore, for future use of the model, we must scollect a large number of data sets for retraining, or choose a more effective target detection model to improve the robustness of the algorithm, as the research in the field of target detection evolves.

## Figures and Tables

**Figure 1 sensors-22-02021-f001:**
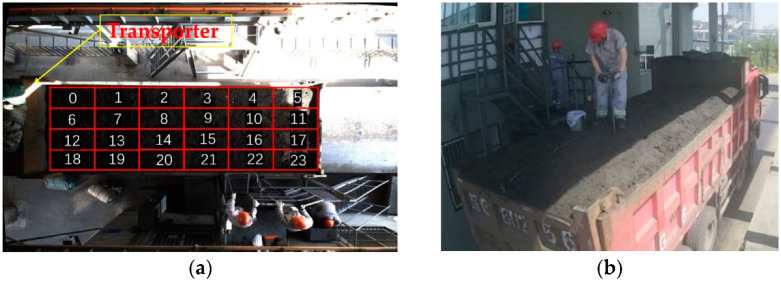
Manual random sampling. (**a**) Sampling area division. (**b**) Manual sampling.

**Figure 2 sensors-22-02021-f002:**
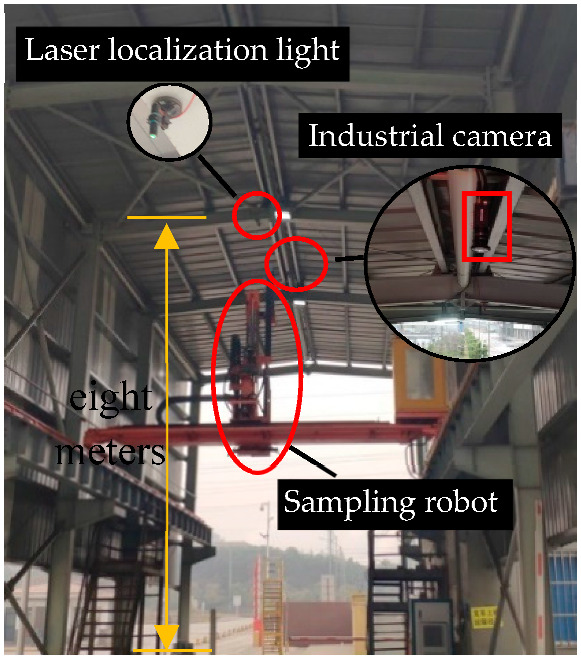
Wide-angle lens shooting.

**Figure 3 sensors-22-02021-f003:**
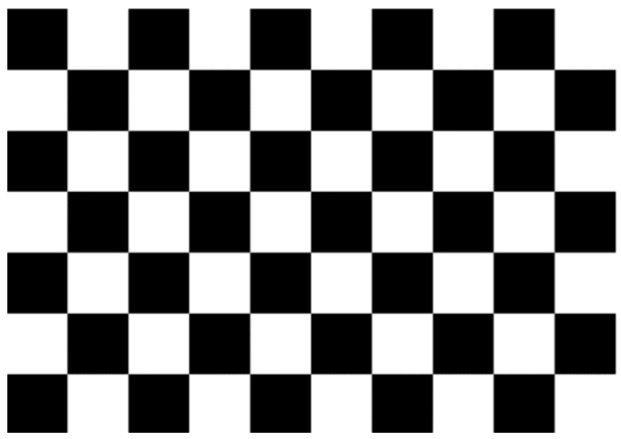
Black and white checkerboard calibration plate.

**Figure 4 sensors-22-02021-f004:**
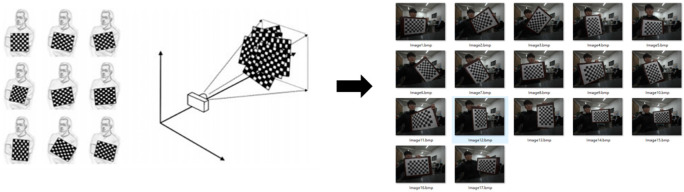
The 17 images were taken by changing the position and angle of the calibration plate.

**Figure 5 sensors-22-02021-f005:**
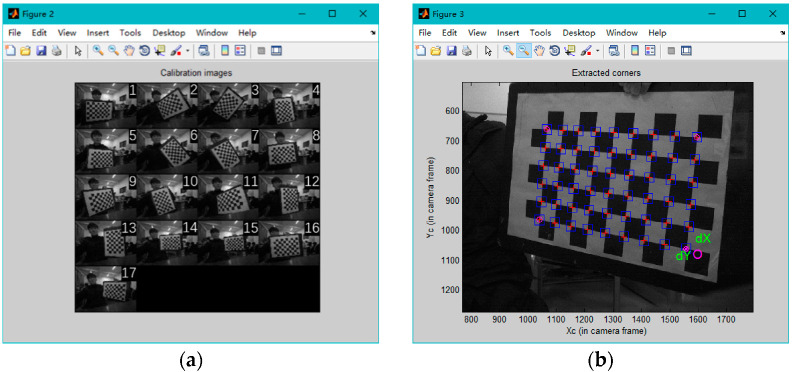
The 17 images loaded by MATLAB and the automatically extracted corners. (**a**) Calibration images. (**b**) Extract corners on the calibration plate.

**Figure 6 sensors-22-02021-f006:**
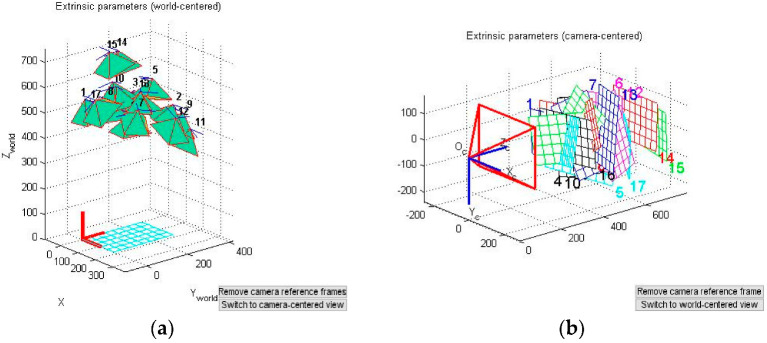
The relative positions and angles of the camera and the calibration plate. (**a**) Camera coordinate system. (**b**) World coordinate system.

**Figure 7 sensors-22-02021-f007:**
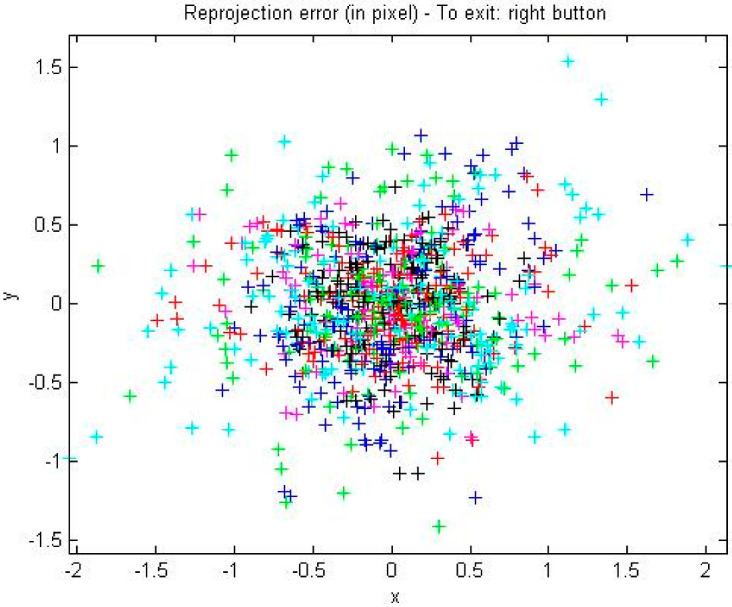
Error distribution diagram of corners.

**Figure 8 sensors-22-02021-f008:**
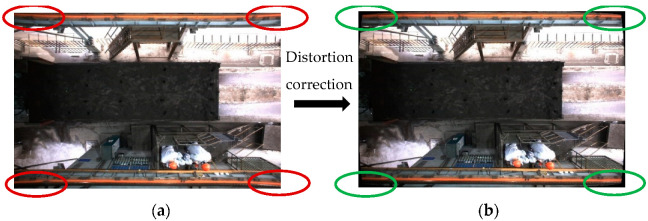
Comparison of before and after distortion correction. (**a**) Before distortion correction. (**b**) After distortion correction.

**Figure 9 sensors-22-02021-f009:**
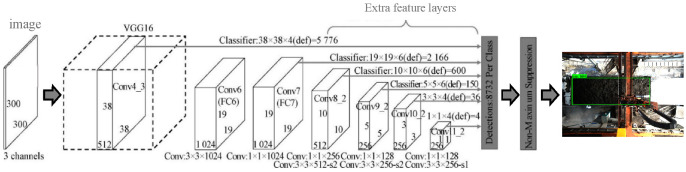
Framework of SSD.

**Figure 10 sensors-22-02021-f010:**
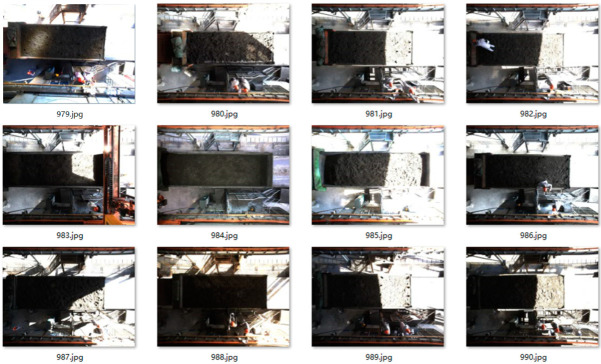
Some training set pictures.

**Figure 11 sensors-22-02021-f011:**
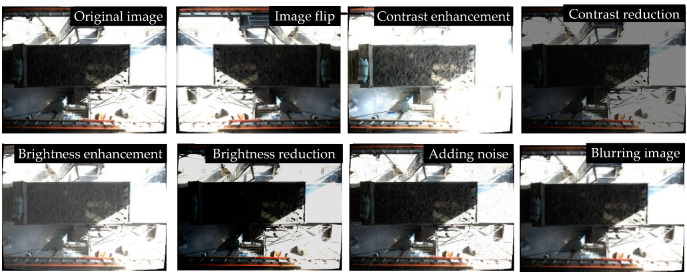
Data augmentation renderings.

**Figure 12 sensors-22-02021-f012:**
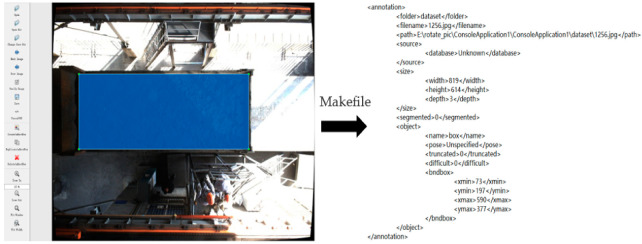
LabelImg data set annotation.

**Figure 13 sensors-22-02021-f013:**
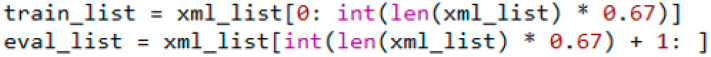
Data set division code.

**Figure 14 sensors-22-02021-f014:**
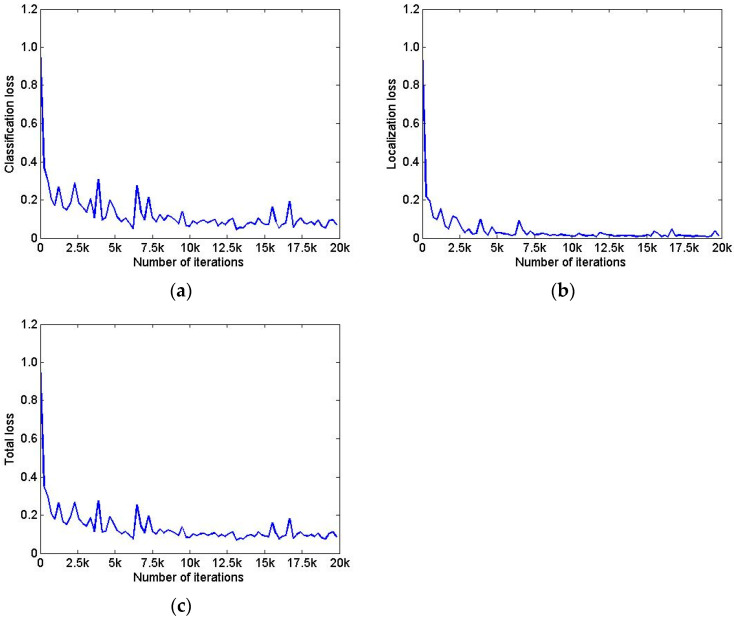
Changing diagram of each loss during model training. (**a**) Changing process of classification loss. (**b**) Changing process of location loss. (**c**) Changing process of total loss.

**Figure 15 sensors-22-02021-f015:**
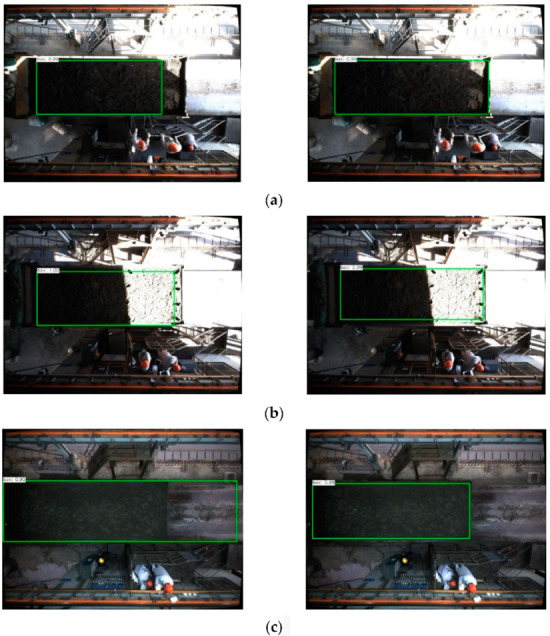
Graphs of detection results in test sets of different scenarios. (**a**) Small part of the lighting scenes. (**b**) Most lighting scenes. (**c**) Cloudy scenes. (**d**) Night lighting scenes.

**Figure 16 sensors-22-02021-f016:**
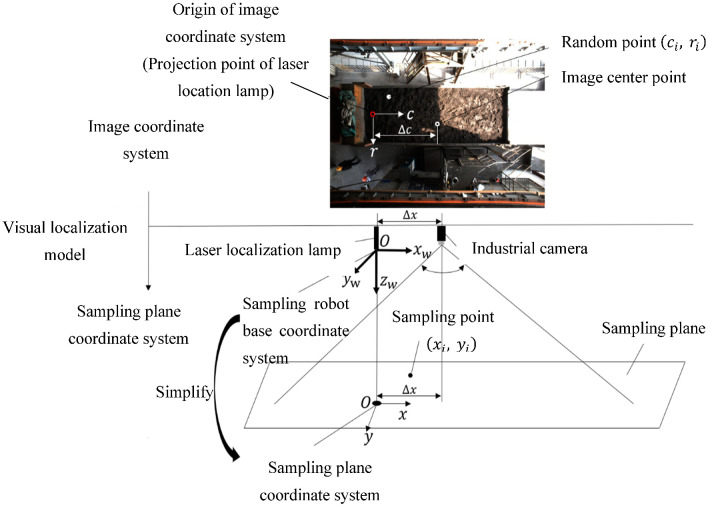
Schematic diagram of the visual localization model.

**Figure 17 sensors-22-02021-f017:**
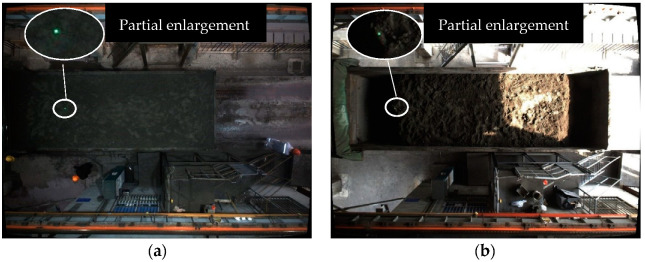
The imaging diagram of the laser spot on the mineral powder plane. (**a**) Cloudy day. (**b**) Sunny day.

**Figure 18 sensors-22-02021-f018:**
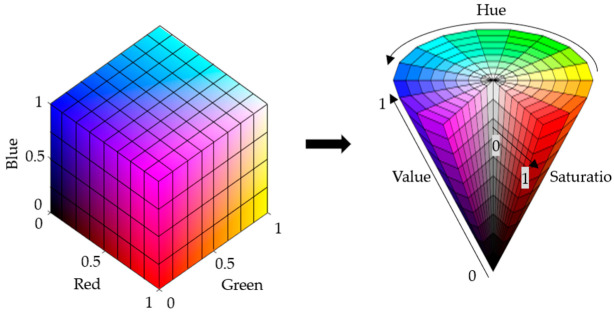
RGB color space to HSV color space conversion.

**Figure 19 sensors-22-02021-f019:**
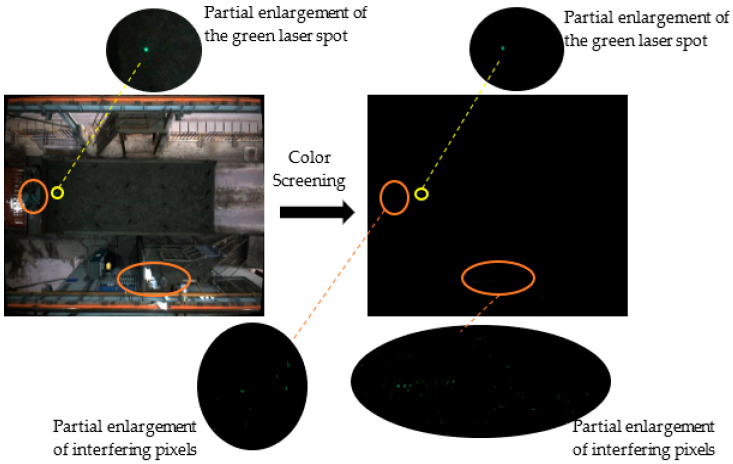
The original image after color screening.

**Figure 20 sensors-22-02021-f020:**
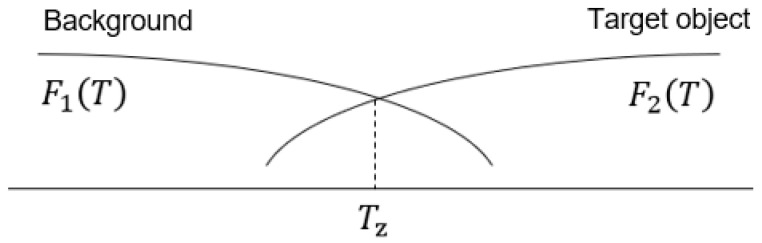
Grayscale value distribution of background and target object.

**Figure 21 sensors-22-02021-f021:**
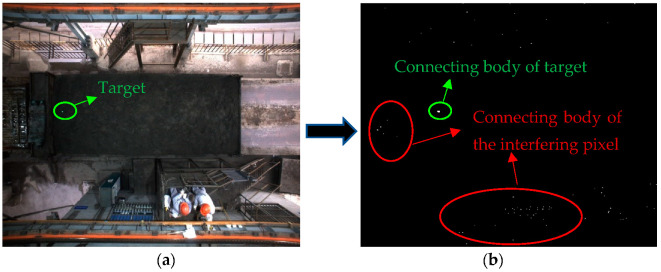
Binary image obtained via the maximum interclass variance method. (**a**) Original image. (**b**) Binary image.

**Figure 22 sensors-22-02021-f022:**
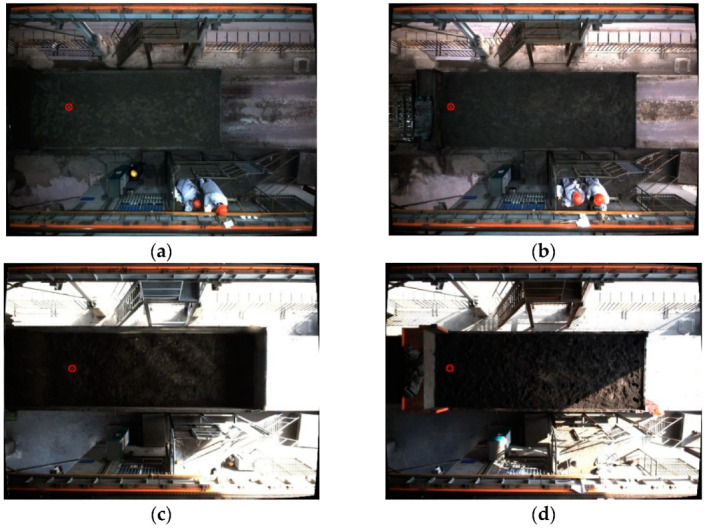
Detection results of green laser spot in different scenes. (**a**) Night scene. (**b**) Rainy day scene. (**c**) Part of the light scene. (**d**) Strong light exposure scenes.

**Figure 23 sensors-22-02021-f023:**
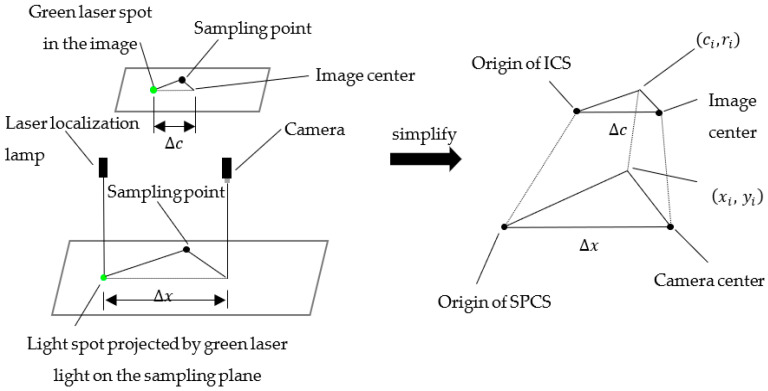
Coordinate conversion schematic.

**Figure 24 sensors-22-02021-f024:**
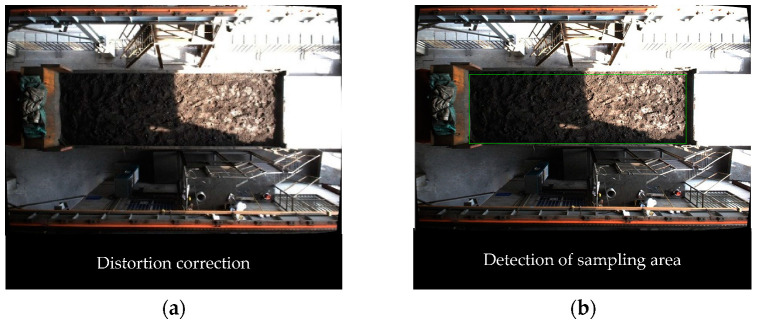
Experimental flow chart.

**Figure 25 sensors-22-02021-f025:**
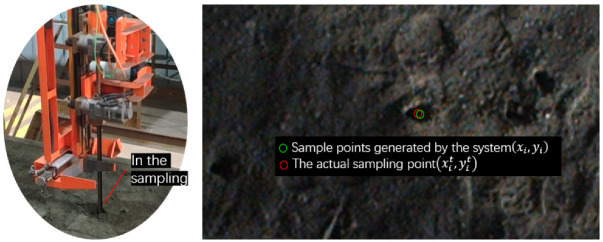
The concave column left after the drill rod is sampled.

**Figure 26 sensors-22-02021-f026:**
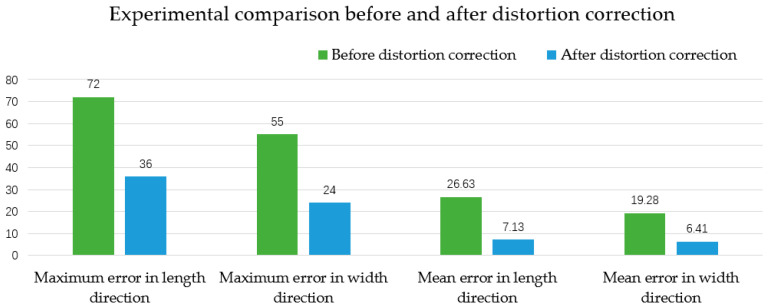
Comparison of before and after distortion correction.

**Figure 27 sensors-22-02021-f027:**
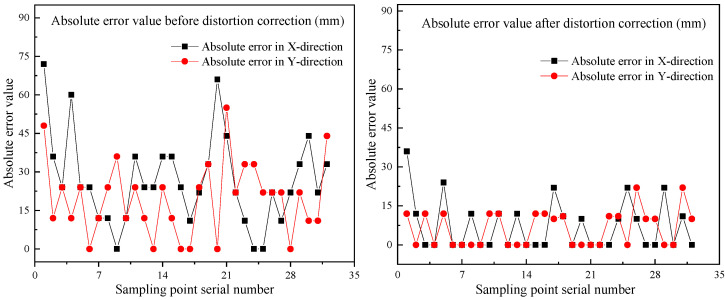
Absolute error in the x-direction and y-direction before and after distortion correction.

**Table 1 sensors-22-02021-t001:** Camera internal parameters table.

Camera Internal Parameters	x-Direction	y-Direction
Focal Length (fc)	1167.20744	1166.56816
±∆fc	±7.30839	±7.06835
Principle Point (cc)	1017.81138	773.90719
±∆cc	±6.56265	±6.01459
$\mathrm{Skew}\;(\alpha_{c})$	0.0000	Angle of Pixel Axes = 90.00000
$\pm \alpha_{c}$	±0.0000	±0.00000

**Table 2 sensors-22-02021-t002:** Distortion parameters table.

Distortion Parameter	$k_{1}\;$	$k_{2}\;$	$k_{3}\;$	$p_{1}\;$	$p_{2}\;$
$\mathrm{Distortion}\;(k_{c\;}\mathrm{and}\;p_{c}$)	−0.11795	0.09822	0.00000	0.00074	−0.00124
$\pm \Delta k_{c}\;\mathrm{and}\;\Delta p_{c}$	±0.01000	±0.02870	±0.00000	±0.00111	±0.00116

**Table 3 sensors-22-02021-t003:** Parameter settings of model training.

Parameter Name	Numerical Value	Parameter Name	Numerical Value
weight_decay	0.0005	decay_factor	0.95
optimizer	Adam	batch_size	8
initial_learning_rate	0.004	num_steps	20,000

**Table 4 sensors-22-02021-t004:** Comparison of SSD model’s detections in 200 test set images.

Sampling Range Detection Situation	Without Data Augmentation	With Data Augmentation
$IoU_{G}\;$< 0.8	23 sheets	7 sheets
Prediction box out of sampling range	25 sheets	6 sheets
No sampling range detected	4 sheets	3 sheets
$0.8\;\leq IoU_{G}\leq 1$ and not exceeding the sampling range	148 sheets	184 sheets
Success ratio	74%	92%

**Table 5 sensors-22-02021-t005:** The setting range of the three HSV channels.

Color Channel	Setting Range
H Channel (Hue Channel)	(55, 77)
S Channel (Saturation Channel)	(43, 255)
V Channel (Value Channel)	(46, 255)

**Table 6 sensors-22-02021-t006:** ICS (in pixels) conversion to SPCS (in mm).

Number	$\mathrm{ICS}\;\left(c_{i},\;r_{i}\right)$	Sampling Plane $\mathrm{Coordinate}\;\mathrm{System}\;\left(x_{i},\;y_{i}\right)$	Number	$\mathrm{ICS}\;\left(c_{i},\;r_{i}\right)$	Sampling Plane $\mathrm{Coordinate}\;\mathrm{System}\;\left(x_{i},\;y_{i}\right)$
1	(458, 65)	(5463, 775)	9	(220, 81)	(2624, 966)
2	(428, 33)	(5105, 394)	10	(220, 1)	(2624, 12)
3	(428, −62)	(5105, −739)	11	(190, −78)	(2266, −929)
4	(399, 65)	(4759, 775)	12	(160, 33)	(1909, 394)
5	(339, 49)	(4044, 584)	13	(131, −46)	(1563, −548)
6	(310, 65)	(3698, 775)	14	(101, 65)	(1205, 775)
7	(280, −62)	(3340, −739)	15	(71, 17)	(847, 203)
8	(250, 33)	(2982, 394)	16	(12, 17)	(143, 203)

**Table 7 sensors-22-02021-t007:** Visual localization results and actual location of the sampled brazing rod (Unit: mm).

Number	Sampling Plane $\mathrm{Coordinate}\;\mathrm{System}\;\left(x_{i},\;y_{i}\right)$	$\mathrm{Real}\;\mathrm{Coordinate}\;\mathrm{System}\;\left(x_{i}^{t},\;y_{i}^{t}\right)$	Absolute Error	Number	Sampling Plane $\mathrm{Coordinate}\;\mathrm{System}\;\left(x_{i},\;y_{i}\right)$	$\mathrm{Real}\;\mathrm{Coordinate}\;\mathrm{System}\;\left(x_{i}^{t},\;y_{i}^{t}\right)$	Absolute Error
1	(5463, 775)	(5499, 787)	(36, 12)	9	(2624, 966)	(2624, 966)	(0, 0)
2	(5105, 394)	(5117, 394)	(12, 0)	10	(2624, 12)	(2624, 0)	(0, 12)
3	(5105, −739)	(5105, −751)	(0, 12)	11	(2266, −929)	(2254, −917)	(12, 12)
4	(4759, 775)	(4759, 775)	(0, 0)	12	(1909, 394)	(1909, 394)	(0, 0)
5	(4044, 584)	(4068, 596)	(24, 12)	13	(1563, −548)	(1575, −548)	(12, 0)
6	(3698, 775)	(3698, 775)	(0, 0)	14	(1205, 775)	(1205, 775)	(0, 0)
7	(3340, −739)	(3340, −739)	(0, 0)	15	(847, 203)	(847, 215)	(0, 12)
8	(2982, 394)	(2994, 394)	(12, 0)	16	(143, 203)	(143, 215)	(0, 12)

**Table 8 sensors-22-02021-t008:** X-direction and y-direction localization error range requirements.

Parameters	X-Direction	Y-Direction
Margin of error	152.78 mm	83.33 mm
Error between actual and localization	≤50 mm	≤50 mm

## Data Availability

Not applicable.

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
