# Peer review of "Intelligent Localization Sampling System Based on Deep Learning and Image Processing Technology"

_sensors, 2022, doi:10.3390/s22052021_

Round 1
Reviewer 1 Report
1) More Citations need to be provided for the statement “powder sampling machines have shortcom-
ings, such as their small sampling range, insufficient intelligent working ability, long
sampling and localization time, and insufficient sampling localization accuracy. ”
2) Acronyms are necessary, for example Single Shot MultiBox Detector (SSD)
3) We feel an improved flow-chart can be generated to get a better understanding of the process (Fig. ?) and a better explicative caption.
4) change 8 meters to eight meters (It is more standard), there are other examples
5) “Based on the ideal pinhole imaging model, the accurate transformation of the coordinate system cannot be realized.” Have you ever think about the stitching of multiple images to get the 360? Or the camera setup is too restrictive?
6) “Color space conversion”Although the use of HSV can help in the detection problem, something that have me worried is the fact that you need hard coded values. Basically any extreme change of illumination can throw change drastically the values being used, Can you explain why only use those values?
7) We find that “Image segmentation” the use of the distribution and the equation is quite interesting given that we have used similar methods for ocr recognition in an industrial setup. However, this depend a lot of getting the correct identification of background and target, we feel this is a little bit weak.
8) Although the use of SSD is used to to find the rectangular slab so sampling can be executed we find this paper more an application than anything else. Can you be more precise in the novelty of it?
Reviewer 2 Report
This manuscript introduces the SSD algorithm based on deep learning is combined with image processing technologies such as color screening and image segmentation to achieve accurate localization and coordinate transformation of sampling points. In summary, the research is interesting and provides valuable results, but the current document has several weaknesses that must be strengthened in order to obtain a documentary result that is equal to the value of the publication.
(1) The abstract is complete and well-structured and explains the contents of the document very well. Nonetheless, the part relating to the results could provide numerical indicators obtained in the research.
(2) The first paragraph introducing the research topic may present a much broad and comprehensive view of the problems related to your topic with citations to authority references.
Y., Tang; M., Zhu; Z., Chen; C., Wu; B., Chen; C., Li; L., Li. Seismic Performance Evaluation of Recycled aggregate Concrete-filled Steel tubular Columns with field strain detected via a novel mark-free vision method. Structures, 2022, 37: 426-441.
F., Wu; J., Duan; S., Chen; Y., Ye; P., Ai; Z. Yang. Multi-Target Recognition of Bananas and Automatic Positioning for the Inflorescence Axis Cutting Point. Frontiers in Plant Science 2021, 12:705021.
(3) In the introduction, the author mentioned the shortcomings of the currently used mine powder sampling machines, such as long sampling and localization time and insufficient sampling localization accuracy. But the author did not explain the performance of the paper’s research method in these aspects.
(4) It is suggested that the author check the article carefully to avoid some mistakes (e.g., in section 2.2.1: “SSD algorithm” instead of “SD algorithm”).
(5) The introduction and proposed system may present a much broad and comprehensive view of the problems. For calibration, please refer to High-accuracy multi-camera reconstruction enhanced by adaptive point cloud correction algorithm; For deformation detection, please refer to Binocular vision measurement and its application in full-field convex deformation of concrete-filled steel tubular columns.
(6) The author processed the image with the maximum interclass variance method to obtain the binary image, which contained the connecting body of the target and the connecting body of the interfering pixel. Subsequently, connecting body feature was screened, and the threshold of screening was determined manually. Is there any guarantee that only the connecting body of the target will be returned after screening?
(7) In order to realize the robot sampling at the target point, after realizing the conversion from the image coordinate system to the sampling plane coordinate system. Do you need to convert the point to robot coordinates?
(8) Data augmentation to ensure the accuracy of SSD algorithm detection is not innovative, and if the latest YOLOv5 network is used for detection, would the detection results be better?
The author failed to compare the research method in this paper with other existing latest technologies, which cannot highlight the superiority of the method in this paper.
Round 2
Reviewer 2 Report
Accept